# The Impact of Emotion Regulation Strategies on Teachers’ Well-Being and Positive Emotions: A Meta-Analysis

**DOI:** 10.3390/bs15030342

**Published:** 2025-03-11

**Authors:** Yi Wang, Fengyu Zai, Xiaoyong Zhou

**Affiliations:** 1Department of General Education, Chuzhou City Vocational College, Chuzhou 239000, China; wangyi@czcvc.edu.cn; 2School of Foreign Languages, East China Normal University, Shanghai 200241, China; 52280400007@stu.ecnu.edu.cn

**Keywords:** emotion regulation strategy, teachers’ well-being, positive emotions, meta-analysis

## Abstract

This meta-analysis investigated the effects of emotion regulation strategies on teachers’ well-being and positive emotional outcomes. Employing a comprehensive review and quantitative synthesis of the literature, the study confirmed that effective emotional regulation, particularly through strategies like deep acting, significantly enhances teacher well-being and job satisfaction. In contrast, surface acting was associated with mixed outcomes, occasionally beneficial but generally not supportive of effective teaching practices. The findings highlight the importance of tailored emotional regulation training in professional development programs for educators. Future research should expand on these findings by exploring diverse emotion regulation strategies across varied educational and cultural contexts to fully understand their impact on educational outcomes.

## 1. Introduction

Teaching is an emotionally demanding profession, and this increasingly challenging landscape is reflected in the significant rise in teacher burnout and attrition rates. Recent studies, including that of [39] ([39]), reveal a 25% increase in burnout rates among educators over the past decade, primarily driven by factors such as the challenges associated with remote learning, student mental health issues, and the diverse needs within classrooms. These intensifying emotional demands highlight the urgent need for effective emotion regulation strategies aimed at fostering teacher well-being and enhancing retention rates. Conversely, disruptive student behaviors often provoke frustration or anger, and self-doubt or perceived inadequacies can lead to anxiety. Furthermore, challenges in communicating concepts effectively may induce frustration. These emotions significantly impact both the educators’ well-being and the quality of instruction, ultimately affecting learner achievement ([3], [4]; [10], [9]). In response to these challenges, teachers often engage in emotional regulation, managing the occurrence, experience, and expression of emotions to maintain professional effectiveness and personal well-being ([15]).

According to the theory of emotional labor, effective teaching requires educators to regulate their emotional displays in accordance with professional norms ([11]). This necessitates displaying positive emotions, such as happiness, while suppressing negative emotions, such as irritation, thereby maintaining a balanced emotional presentation ([31]; [36]; [43]). Specifically, teachers may employ deep acting strategies, aligning their internal emotional state with their outward expressions, or surface acting, whereby they modulate their authentic emotions or simulate those not genuinely experienced ([12]; [14]). The essential role of emotional regulation in educational settings is underscored by how these strategies contribute to fostering positive classroom dynamics and supporting effective student learning.

Empirical research has highlighted the significant impact of emotion regulation strategies on teachers’ professional experiences. Studies have shown that the type of emotional labor in which teachers engage affects their commitment, self-esteem, job satisfaction, and susceptibility to burnout ([18]; [20]; [24]). For instance, [24] ([24]) identified a positive correlation between emotional exhaustion and both high emotional job demands and frequent use of surface acting. Conversely, [43] ([43]) and [47] ([47]) found positive relationships between deep acting and job satisfaction, though [5] ([5]) reported no significant connection.

Despite growing interest in teacher emotion regulation, significant gaps remain in understanding how specific strategies, such as deep acting versus surface acting, differentially impact teacher well-being and retention. This study addresses this gap by providing a comprehensive meta-analysis of these strategies and their implications for contemporary educational challenges, such as rising teacher attrition rates and the emotional complexities of modern classrooms. The research has identified two primary objectives of teachers’ emotion regulation: enhancing teaching effectiveness and conforming to the ideal emotional demeanor expected of teachers ([32]).

This study builds on [12]’s ([12]) model and utilizes meta-analyses to assess the effectiveness of teacher emotion regulation strategies comprehensively, examining a broad array of antecedents and outcomes. Through the analysis of qualitative data on teacher emotion, we explore the strategies teachers utilize in classrooms. This research aims to examine the relationships between teachers’ emotion regulation strategies and various factors, including emotional labor, job satisfaction, psychological traits, teaching efficacy, self-consciousness, social belongingness, and language and creativity. Additionally, it seeks to determine if cultural, conceptual, or grade-level differences moderate these relationships, providing a more nuanced understanding of how emotion regulation contributes to educational success.

## 2. Literature Review

### 2.1. Emotion and Emotion Regulation

Emotions play a fundamental role in the teaching process, with recent research emphasizing their critical influence on teacher well-being and classroom effectiveness ([39]). In contemporary educational settings, educators encounter increased emotional demands stemming from factors such as remote learning, student mental health issues, and the diverse needs of their classrooms ([42]). These challenges underscore the necessity of effective emotion regulation strategies to enhance teacher resilience and job satisfaction. [28] ([28]) assert that emotions not only arise from cognitive processes but also substantially shape these processes and motivation. For instance, negative emotions have been shown to impair working memory, a critical system for storing and processing information during various cognitive tasks ([28]). Conversely, engaging the working memory capacity can mitigate negative emotional states ([37]). Additionally, positive emotions are known to expand cognitive flexibility and action repertoire, suggesting that students and teachers experiencing positive emotions are likely to develop more innovative ideas and strategies ([8]). This interplay between emotions and cognition highlights the profound importance of managing emotions in educational settings, which not only prevents cognitive impairment but also enhances the overall learning experience, thus emphasizing the need for training educators in emotional intelligence.

Furthermore, emotions substantially influence cognitive, regulatory, and problem-solving strategies ([25]). Specifically, negative emotions can inhibit students’ utilization of deep cognitive strategies that necessitate thorough and intricate information processing ([28]), while compelling teachers to adopt inflexible and uncreative instructional approaches. Emotions serve a pivotal function in how teachers and students process information and approach problem-solving tasks ([34]). Empirical evidence demonstrates that positive emotions significantly enhance intrinsic motivation, illustrating the profound impact emotions exert on both cognitive engagement and motivational processes ([28]). Despite their fundamental importance, educational psychology and language learning research have traditionally emphasized learner emotions and their regulation while giving insufficient attention to teachers’ emotional experiences and regulatory strategies ([25]; [41]; [40]). This imbalance underscores the necessity for a more comprehensive approach that adequately addresses teachers’ emotional regulation, recognizing that their emotional well-being is essential for cultivating an effective and supportive learning environment.

Existing research illuminates the diverse ways teachers regulate their emotions within the classroom context, although considerable uncertainties persist regarding the specific motivations and methodologies employed. The primary emphasis has been on emotional labor strategies, particularly deep and surface acting ([12]). Deep acting involves the internalization of desired emotions to achieve congruence between expressed emotions and authentic feelings, typically accomplished through cognitive techniques such as distraction or self-persuasion. Conversely, surface acting entails the manifestation of inauthentic emotions through either simulation or suppression of genuine feelings ([12]). Research has consistently demonstrated the significance of implementing appropriate emotion regulation strategies, as ineffective emotional management among educators frequently results in burnout, job dissatisfaction, and diminished teaching effectiveness ([19]; [45]). This evidence emphasizes the imperative for targeted emotional regulation interventions that transcend merely addressing symptoms of emotional misalignment to cultivate authentic emotional resilience, thereby enabling educators to flourish in their demanding professional environment.

Further research indicates that teachers engage in emotional regulation because they believe it enhances their teaching efficacy and improves classroom management and student relationships ([31]; [33]). Teachers also consider emotional regulation integral to their professional roles, enhancing teaching effectiveness and fostering better student relationships. Recent empirical studies further highlight the advantages of emotional regulation for teachers, linking it to increased job commitment and enhanced psychological well-being—both of which are crucial for the success and longevity of language teachers’ careers ([38]; [2]). Effective emotional regulation allows language teachers to engage more profoundly in their teaching, applying both internal and external processes to manage their emotions and achieve specific educational goals.

### 2.2. Emotion Regulation Strategies

Emotional regulation is accomplished through various strategies. Extensive research has identified the consequences of neglecting teachers’ emotional regulation, including burnout, job dissatisfaction, inefficiencies in teaching and classroom management, and increased attrition rates ([23]). The range of emotion regulation strategies is broad, with various theoretical frameworks offering different categorizations. For example, Gross categorizes emotion regulation strategies into reappraisal and suppression, based on the timing of the regulation relative to the emotion’s onset ([16], [17]). Reappraisal involves altering one’s emotional response by reinterpreting a situation, while suppression involves inhibiting the outward expression of emotions.

Additionally, educational theorists such as [29] ([29]) have introduced strategies designed to foster positive student engagement through emotional regulation in the classroom. These strategies include enhancing curiosity, interest, and pleasure, as well as managing anxiety. Other scholars, such as [35] ([35]), advocate for a wide range of teaching-related emotion regulation strategies, from psychological matching to communication and avoidance, emphasizing the need for both individual and collective emotional management in educational settings.

Grandy’s emotional labor model (2000) links reappraisal with deep acting and suppression with surface acting. Deep acting aligns one’s internal emotions with external expressions, fostering genuine emotional displays, whereas surface acting involves modifying outward emotional expressions without altering internal feelings, often required by professional roles that demand constant interaction, such as teaching or customer service ([12]). In her later work, Grandey redefines emotional labor as a form of emotional regulation, encouraging further exploration of emotion regulation strategies beyond the traditional dichotomy of surface and deep acting ([13]). This conceptualization offers a comprehensive framework for understanding the antecedents and consequences of emotional regulation in the teaching profession. Examining emotion regulation strategies, such as deep and surface acting, is crucial, as these strategies are theorized to influence various outcomes, including emotional labor, job satisfaction, self-efficacy, resilience, emotional intelligence, and even creativity among teachers.

While a substantial body of research underscores the importance of emotion regulation within teaching and learning contexts, notable gaps persist—particularly concerning the combined effects of various emotion regulation strategies on teacher well-being and the cultivation of positive emotions. Furthermore, prior studies have primarily concentrated on individual case studies and qualitative assessments, lacking a systematic, quantitative overview that could facilitate more generalizable findings. This study seeks to address these deficiencies through a comprehensive meta-analysis that assesses the impact of different emotion regulation strategies on teacher well-being and the enhancement of positive emotions. By illuminating the most effective strategies, this analysis aims to deepen our understanding of the mechanisms by which emotion regulation influences teachers’ emotional states and professional well-being in educational settings. Ultimately, this research endeavors to fill the existing gap by examining the specific effects of diverse emotion regulation strategies on teachers’ positive emotions, thereby contributing to a more nuanced comprehension of the factors that promote effective teaching.

## 3. Methodology

### 3.1. Literature Search

This study followed the PRISMA protocol for systematic reviews and meta-analyses. The research process was divided into three phases: (1) systematic search and identification, (2) screening and eligibility assessment, and (3) data extraction and quality assessment. The electronic search was performed across four major databases: WOS, EBSCO, JSTOR, and ProQuest. In addition, we also selected the Chinese databases CNKI and WEIPU. These databases were selected to ensure comprehensive coverage of relevant literature. The electronic search was conducted using Boolean logic to combine keywords and phrases. The exact search terms included (“emotion regulation” OR “emotional labor”) AND (“teacher emotion regulation strategies” OR “educator emotion regulation”) AND (“emotional labor” OR “job satisfaction” OR “well-being” OR “achievement”). For example, the search string in Web of Science was: TS = ((“emotion regulation” OR “emotional labor”) AND (“teacher emotion regulation strategies” OR “educator emotion regulation”) AND (“emotional labor” OR “job satisfaction” OR “social belonging” OR “well-being” OR “achievement”)). Similar search strings were adapted for other databases, such as EBSCO, JSTOR, and ProQuest, to ensure consistency and comprehensiveness. In Chinese databases (CNKI and WEI PU), we used terms such as “教师情绪调节策略” (teacher emotion regulation strategies) and “教师情绪” (teacher emotion). The search was limited to the literature published in the past two decades.

### 3.2. Inclusion and Exclusion Criteria

#### 3.2.1. Inclusion Criteria

Studies that met our inclusion criteria were initially assessed through a title and abstract review, incorporating all studies that contained the specified terms. To be eligible for inclusion, papers had to meet the following criteria:Studies concentrating on in-service teachers at various stages and in different countries. For instance, “*Choose your strategy wisely: Examining the relationships between emotional labor in teaching and teacher efficacy in Hong Kong primary schools*” focuses on a sample of 1115 Hong Kong primary school teachers ([44]).Studies concentrating on the emotion regulation strategies for regulating teachers’ emotions. For instance, “*Adaptation and validation of the teacher emotional labor strategy scale in China*”. This article reports the adaptation and validation of the Teacher Emotional Labor Strategy Scale (TELSS) as tested on samples of 633 Beijing teachers and 648 Chongqing teachers in the Chinese mainland ([43]).A quantitative study that assesses teacher emotional regulation by employing reliable, adequate, and effective data sample size. For instance, “*Emotional intelligence, emotional labor strategies and satisfaction of secondary teachers in Pakistan*”. In this article, there is a sufficient and effective amount of data.The included studies are obliged to explore the relationship between emotion regulation strategies and teachers’ emotional regulation. For instance, “*Spiral effects of teachers’ emotions and emotion regulation strategies: Evidence from a daily diary study*”. In this article, it is expected that positive emotions will promote teachers’ use of adaptive emotion regulation strategies ([21]).The full text of the article is published in English. For instance, “*The associations between EFL learners’ L2 class belongingness, emotion regulation strategies, and perceived L2 proficiency in an online learning context*”.

#### 3.2.2. Exclusion Criteria

To further refine our selection, we have established specific exclusion criteria to guarantee that our review includes only the most relevant and high-quality studies:Exclude participants who may not have classroom teaching experience, such as intern teachers, teaching assistants, and school counselors. For instance, in the article “*Positive Influence of Cooperative Learning and Emotion Regulation on EFL Learners’ Foreign Language Enjoyment*”, the participants are EFL Learners’.The study also excluded those emotion regulation strategies that did not focus on the teacher’s emotions, such as “emotional regulation ability”, and also excluded studies that focused on the “teacher’s negative emotions”, as this was exactly opposite to the study’s focus on “positive emotions”. For instance, in “*Linking emotion regulation strategies to affective events and negative emotions at work*” the main content of the article is how to use emotion regulation strategies to alleviate teachers’ negative emotions.Exclude studies with incomplete data or unclear outcome effects to ensure the completeness and accuracy of the analysis. For instance, “*Teachers’ Emotion Regulation and Classroom Management*”. Despite the fact that the content of this article is relevant, there is not enough data.Articles with weak relevance to important information related to the research variables, excluding articles in languages other than English.There are no quantitative components in the study (such as interview reports, focus group reports, classroom observation reports), for instance, “*Measuring Language Teacher Emotion Regulation: Development and Validation of the Language Teacher Emotion Regulation Inventory at Workplace*”.

### 3.3. Selected Studies

The literature selection process for the meta-analysis was systematically conducted, as illustrated in the flowchart. Initially, a comprehensive search was carried out across both English and Chinese databases, including Web of Science, EBSCO, JSTOR, ProQuest, CNKI, and WEI PU, yielding a total of 4278 records. After removing duplicates, the remaining records totaled 4213, which were screened based on titles and abstracts. The screening process resulted in the exclusion of a substantial number of records, with 4148 entries being dismissed due to a lack of relevance, leaving 65 studies initially deemed eligible for further analysis.

These 65 studies were subjected to a detailed examination, with 53 being excluded due to missing data critical for the meta-analysis, resulting in 12 studies being included in the quantitative synthesis. This final selection was based on rigorous criteria to ensure the relevance and quality of the data for contributing to the meta-analysis objectives, aiming to assess the impact of emotion regulation strategies on teachers’ well-being and positive emotions (see Figure 1).

### 3.4. Coding

The author coded the included articles with (a) No., (b) Publication date, (c) Journal name, (d) Sample size, (e) Participants, (f) Emotion regulation strategies, (g) Teachers’ emotions, (h) Correlation coefficient value (see Table 1). After initial coding, the author developed a comprehensive list that categorized the articles and their research variables into the eight corresponding factors. Here is a list of these factors, with more detailed information available in Table 1.

### 3.5. Statistical Analysis

All data were analyzed using Comprehensive Meta-Analysis Software V 3.7 (CMA) to merge effect sizes and analyze moderated effects. The meta-analysis employed the [6] ([6]) technique. A heterogeneity test assesses the variability in study outcomes. It evaluates whether the studies measure the same underlying effect or if significant differences exist, which may require further investigation. In this research, the Q value test and the I^2^ statistic are used to assess heterogeneity. The Q value indicates heterogeneity, while I^2^ represents the proportion of variance in the effect size due to heterogeneity. An I^2^ of 0 indicates homogeneity. Values between 0 and 40% suggest mild heterogeneity, 40–60% moderate heterogeneity, 50–90% considerable heterogeneity, and 75–100% substantial heterogeneity.

The heterogeneity test results (see Table 2) were significant with a Q-value of 3994.683, indicating substantial heterogeneity (I^2^ = 98.598%, PQ < 0.05), which suggests a strong discrepancy among the included studies. According to the discrimination criterion, when I^2^ exceeds 75%, it indicates strong heterogeneity among the studies ([30]). Given this high degree of heterogeneity, a random-effects model was appropriately employed to provide a more generalized interpretation of the effects across diverse study conditions. This approach acknowledges that the studies included are sampled from populations of studies that potentially differ in substantial ways.

The high heterogeneity observed in our meta-analysis (I^2^ = 998.598%) indicates substantial variability across the included studies. This variability may arise from differences in cultural contexts, educational systems, and the specific emotion regulation strategies employed by teachers. As evidenced in Table 1, the countries of the participants and the educational levels at which they teach vary significantly, which may contribute to this observed heterogeneity. Additionally, the disparities in sample sizes across studies, ranging from a minimum of 62 participants (No. 8) to a maximum of 2022 participants (No. 2), may further exacerbate the high levels of heterogeneity. While this variability limits the generalizability of our findings, it simultaneously underscores the complexity of emotional regulation in teaching and the need for context-specific interventions. Therefore, future research should investigate moderating factors, such as cultural differences and teaching environments, to gain a more nuanced understanding of how these variables influence the effectiveness of emotion regulation strategies.

### 3.6. Publication Bias

Publication bias refers to a phenomenon where studies with more favorable results are more likely to be accepted and published by journals. It is a crucial factor influencing the reliability of research findings, and thus, its examination is an indispensable and important step in meta-analysis. To prevent the deviation of the analysis results caused by publication bias, this article conducted a publication bias test for the effect values of each variable. This article initially employed the funnel plot to make a subjective determination of whether there was a publication bias for each variable. The funnel plot is a common visual method used to detect publication bias in meta-analyses. Ideally, studies should symmetrically distribute around the combined effect size (diamond at the plot’s base), forming an inverted funnel shape. As shown in Figure 2, it can be observed that each variable is uniformly distributed on both sides at the top of the funnel, suggesting that there is no substantial publication bias. To further examine the aforementioned findings, this paper adopts the Egger test approach for quantitative verification. The test results are presented in Figure 3. The Egger test determines whether there is publication bias by identifying whether there is a significant difference between the regression intercept and 0. In general, an Egger test *p* > 0.05 indicates a low likelihood of publication bias. As shown in Figure 3 (T = 0.203, *p* = 0.839), this indicates that the overall meta-analysis is less likely to be affected by publication bias. In addition, we also employed Kendall’s tau method, which is a non-parametric statistical method used to measure the correlation (i.e., degree and direction) between two variables. Kendall’s tau is primarily used to evaluate the degree of association between ordinal data. The rank correlation analysis showed no significant difference (z = 0.165, *p* = 0.868) (see Figure 4).

## 4. Results

### 4.1. Descriptive Findings

The meta-analysis included a total of 57 effect sizes derived from 12 independent studies conducted over the past two decades, as detailed in Table 1. Among these, eight studies focused on the emotion regulation strategies of deep acting and surface acting, two studies explored the process model of emotion regulation, specifically cognitive reappraisal and suppression. Additionally, two studies examined a broader range of emotion regulation strategies. The sample populations predominantly consisted of Chinese teachers, with significant contributions from Croatian, Pakistani, and Turkish educators as well.

This analysis involved an extensive review of the antecedents, such as deep acting, surface acting, reappraisal, and suppression, and their consequent effects on teacher outcomes, including emotional labor, job satisfaction, self-efficacy, and psychological resilience. Through this detailed examination, the study aimed to better understand the complex dynamics of how different emotion regulation strategies impact teachers’ well-being and positive emotions.

### 4.2. Effect Size Analysis

The effect size is a statistical index used to measure the magnitude of the experimental effect or the strength of the relationship between variables. Importantly, it is largely independent of the sample size. Following Cohen’s guidelines for interpreting the strength of correlations: an *r*-value from 0.00 to 0.09 suggests a very weak correlation; from 0.10 to 0.29, a weak correlation; from 0.30 to 0.49, a moderate correlation; and from 0.50 to 1.00, a strong correlation. The results of the effect sizes for all dimensions are displayed in Table 3.

#### 4.2.1. The Effect of Surface Acting on Teachers’ Emotions

The results in Table 3 show that while surface performance occasionally provides minor benefits, it usually does not support or may even undermine aspects that are critical to effective teaching and learning. While there is a negligible negative correlation with emotional and psychological traits (r = −0.131, *p* = 0.099) and emotional labor (r = −0.025, *p* = 0.832), suggesting minimal impact, a significant negative correlation is observed with language and creativity (r = −0.830, *p* < 0.001), indicating that surface acting may substantially hinder creative and linguistic functions. Conversely, a small but statistically significant positive effect is noted in teaching efficacy (r = 0.103, *p* < 0.001), suggesting that surface acting might marginally enhance perceived efficacy in classroom management. However, the overall effect across all outcomes is weak and statistically non-significant (r = −0.076, *p* = 0.280), with extremely high heterogeneity (I^2^ = 99.008), pointing to considerable variability in how surface acting influences different aspects of teaching.

#### 4.2.2. The Effect of Deep Acting on Teachers’ Emotions

The findings presented in Table 3 highlight the influence of deep acting as an emotion regulation strategy on various dimensions of teachers’ professional experiences. In summary, the data presented in Table 3 indicate that while certain strategies, such as deep acting and general emotion regulation, tend to produce positive effects across multiple outcomes, others, such as surface acting and emotional suppression, exhibit mixed or negative effects. This variability in outcomes across different strategies and domains underscores the complexity of emotional regulation within the teaching profession and suggests that the effectiveness of these strategies is highly context-dependent. Specifically, context may encompass factors such as the type of school (e.g., public vs. private), the overall school climate (e.g., supportive vs. stressful), the classroom environment (e.g., collaborative vs. competitive), and the educational level being taught (e.g., primary vs. secondary education). Therefore, future research should investigate how these contextual factors moderate the effectiveness of emotion regulation strategies, thereby providing more tailored recommendations for educators operating in diverse settings.

Specifically, the correlation with emotional and psychological traits was significant (r = 0.215, *p* < 0.001), suggesting that deep acting positively affects teachers’ emotional stability and psychological health. Although the correlation with emotional labor was positive (r = 0.222), it was not statistically significant (*p* = 0.130) and exhibited high heterogeneity (I^2^ = 98.536), indicating substantial variability across studies. Job satisfaction showed a significant positive correlation with deep acting (r = 0.213, *p* < 0.001), highlighting its potential to enhance satisfaction levels among teachers. Similarly, self-consciousness was positively correlated with deep acting (r = 0.195, *p* < 0.001), suggesting that deep acting helps increase awareness of emotional states. The weakest yet significant positive correlation was observed with teaching efficacy (r = 0.140, *p* < 0.001), indicating a modest impact. Overall, the meta-analysis revealed a consistent but weak positive effect of deep acting across the examined outcomes (total r = 0.201, *p* < 0.001), with moderate overall heterogeneity (I^2^ = 95.178), underscoring the nuanced impact of deep acting on teachers’ professional experiences.

#### 4.2.3. The Effect of Emotion Regulation on Teachers’ Emotions

The effects of general emotion regulation on emotional and psychological traits, as well as language and creativity, are inconclusive because with only one piece of literature, there is no way to conduct a systematic analysis. The overall effect is also significant and positive (r = 0.668, *p* < 0.001) with high heterogeneity (I^2^ = 95.921).

#### 4.2.4. The Effect of Reappraisal on Teachers’ Emotions

Reappraisal (see Table 3) shows a positive effect on Job Satisfaction (r = 0.160) and a stronger positive impact on Social Belongingness (r = 0.397, *p* < 0.001). The overall effect of reappraisal across the studied outcomes is positive (r = 0.256, *p* < 0.001) with moderate heterogeneity (I^2^ = 88.080).

#### 4.2.5. The Effect of Suppression on Teachers’ Emotions

The effect of suppression (all details in Table 3) with on job satisfaction could not be determined because the amount of literature was only one. However, there was a positive, albeit small, effect on social belonging (r = 0.156, *p* < 0.001). The overall effect size is negligible and not statistically significant (r = −0.014, *p* = 0.262) with high heterogeneity (I^2^ = 89.601).

In summary, the data from Table 3 indicates that while some strategies like deep acting and general emotion regulation tend to yield positive effects across several outcomes, others like surface acting and suppression exhibit mixed or negative impacts. The variability in effects across different domains and strategies highlights the complexity of emotional regulation within the teaching profession and suggests that the effectiveness of these strategies can be highly context-dependent.

## 5. Discussion

In this meta-analysis, we explored the critical role of emotion regulation in teaching, with a focus on its impact on teachers’ well-being and positive emotions. The reviewed literature highlights that teaching is inherently an emotional endeavor, influenced by the teacher’s ability to manage both personal and professional emotions. Consistent with prior research ([8]; [25]), our findings reaffirm that emotions are integral not only to cognitive processes in teaching but also to motivational and regulatory dynamics within the classroom. Our findings also indicate that deep acting consistently enhances teacher well-being and job satisfaction, aligning with previous studies ([43]; [47]). In contrast, surface acting showed mixed outcomes, with a significant negative impact on creativity and language instruction, suggesting that inauthentic emotional displays may hinder effective teaching in these domains. These results underscore the importance of fostering authentic emotion regulation strategies, such as deep acting, in professional development programs, However, the small but significant positive effect of surface acting on teaching efficacy suggests that, in certain contexts, surface acting may provide short-term benefits in classroom management. This nuanced finding highlights the need for tailored interventions that consider the specific demands of different teaching environments.

This meta-analysis highlights the benefits of deep acting in mitigating burnout and enhancing job satisfaction, particularly in the context of remote learning and increased student mental health concerns ([42]). These findings suggest that professional development programs should prioritize training in deep acting strategies to improve teacher retention and classroom effectiveness, particularly in high-stress educational environments. This finding aligns with studies by [43] ([43]) and [47] ([47]), which documented positive correlations between deep acting and job satisfaction among teachers. These strategies help educators manage classroom interactions and foster a supportive learning environment. In contrast, surface acting—suppressing genuine emotions or displaying inauthentic ones—had more complex outcomes. Although surface acting occasionally contributed to perceived teaching efficacy, it generally failed to support—and could even undermine—critical aspects of teaching, especially in areas demanding high emotional authenticity, such as language instruction and creativity. This is consistent with existing literature, which associates surface acting with negative outcomes, where the dissonance between felt and expressed emotions leads to emotional exhaustion and decreased job satisfaction ([24]). While our study focused primarily on deep acting and surfacing acting, other emotion regulation strategies, such as cognitive reappraisal and suppression, also warrant attention. Cognitive reappraisal, which involves reinterpreting emotional stimuli to alter their impact, has been shown to reduce stress and enhance emotional resilience ([17]). Suppression, on the other hand, involves inhibiting the outward expression of emotions and has been associated with increased physiological stress and reduced job satisfaction ([13]). Future research should explore the effectiveness of these strategies in educational settings, particularly in high-stress environments such as remote learning or classrooms with diverse student needs. Integrating a broader range of emotion regulation strategies into teacher training programs could provide educators with more tools to manage their emotions effectively and improve their overall well-being. A significant contribution of this study is the systematic quantification of the effects of various emotion regulation strategies on teacher outcomes, addressing a gap in the existing literature. While prior research often relied on qualitative assessments or individual case studies, this meta-analysis offers a broader, quantitative overview, providing more generalized insights into the effectiveness of these strategies.

Although our analysis revealed no substantial evidence of publication bias (Egger’s test: T = 0.203, *p* = 0.839), the potential for bias remains a limitation inherent in meta-analytic studies. It is well-established that studies with significant or favorable outcomes are more likely to be published, which can skew the overall effect sizes. To address this concern, we conducted a thorough search across multiple databases, incorporating both English and Chinese sources, and utilized the Egger test and funnel plot analysis to evaluate bias. Despite these precautions, the possibility that unpublished null findings may influence our results cannot be entirely dismissed. Therefore, future research should strive to include gray literature and unpublished studies to further minimize the risk of publication bias.

However, this study has several limitations. Although the meta-analysis consolidated findings from various studies, it focused mainly on deep and surface acting, with less emphasis on other potentially influential strategies, such as cognitive reappraisal or emotional suppression, and their broader educational impact. Future research should expand the range of emotion regulation strategies and examine their effects across diverse educational and cultural contexts. Longitudinal studies could offer deeper insights into the long-term effects of emotion regulation training on teachers’ professional growth and student outcomes. Moreover, incorporating mixed-methods research could enrich quantitative findings with qualitative insights, offering a more comprehensive understanding of the nuances of emotion regulation in education.

In conclusion, this meta-analysis highlights the importance of emotion regulation in teaching, showing its significant effects on teacher well-being and positive emotions. By enhancing our understanding of how emotion regulation strategies affect educational outcomes, this research contributes to developing targeted interventions to foster emotional competencies in teachers, ultimately improving education quality. As the field progresses, adopting a holistic approach to studying emotion in education will be crucial for building more resilient and effective educational systems.

## 6. Conclusions

This meta-analysis underscores the critical role of emotion regulation strategies in enhancing teachers’ well-being and efficacy. Consistent with the existing literature, the findings show that deep acting, in which teachers align their internal feelings with outward expressions, significantly enhances job satisfaction and overall well-being. However, surface acting had mixed effects: it was occasionally beneficial but generally undermined key aspects of effective teaching, especially in contexts where emotional authenticity is critical. These outcomes highlight the complex dynamics of emotional regulation in education and underscore the need for targeted training in emotional regulation within teacher development programs. Future research should explore a broader range of emotion regulation strategies and assess their long-term impacts across diverse educational and cultural contexts to better tailor interventions to specific teaching environments.

## Figures and Tables

**Figure 1 behavsci-15-00342-f001:**
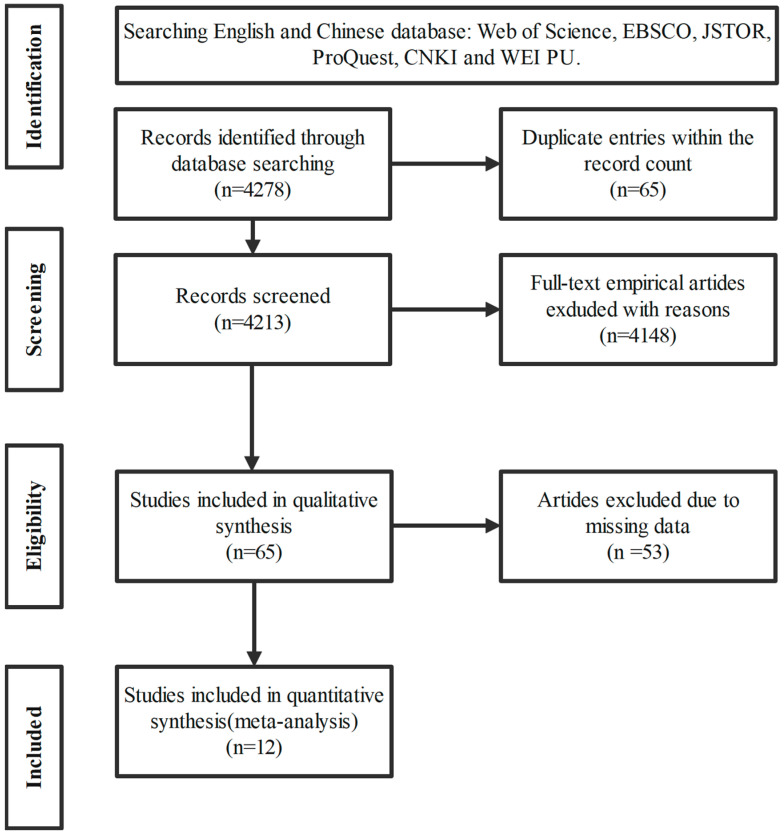
PRISMA flowchart for the identification, screening, and inclusion of publications in the meta-analyses.

**Figure 2 behavsci-15-00342-f002:**
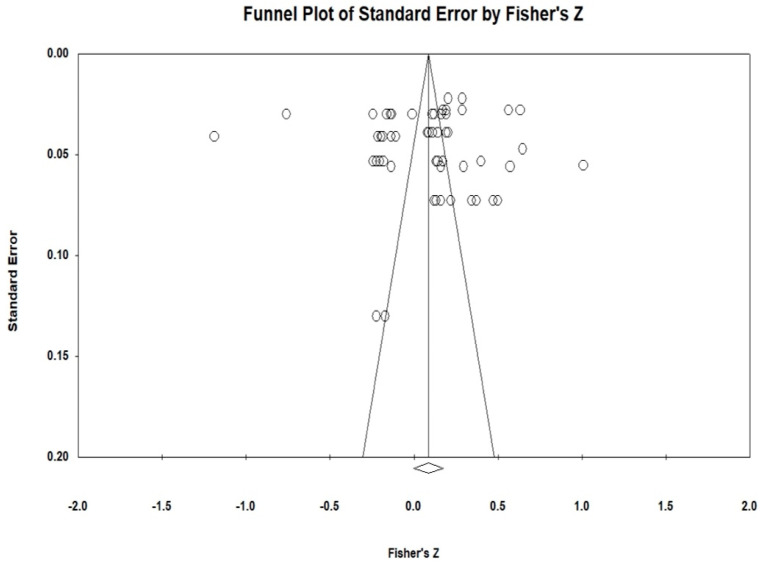
Funnel chart results.

**Figure 3 behavsci-15-00342-f003:**
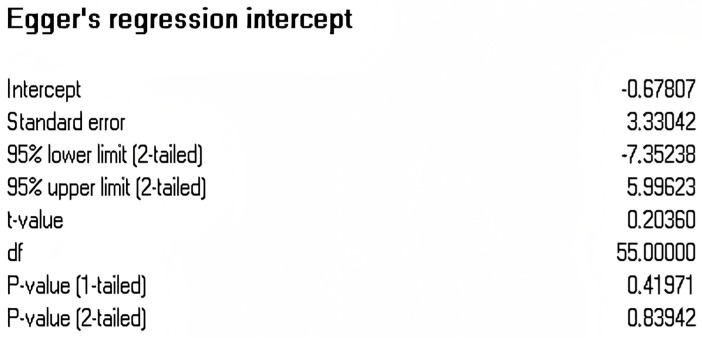
Egger’s regression intercept.

**Figure 4 behavsci-15-00342-f004:**
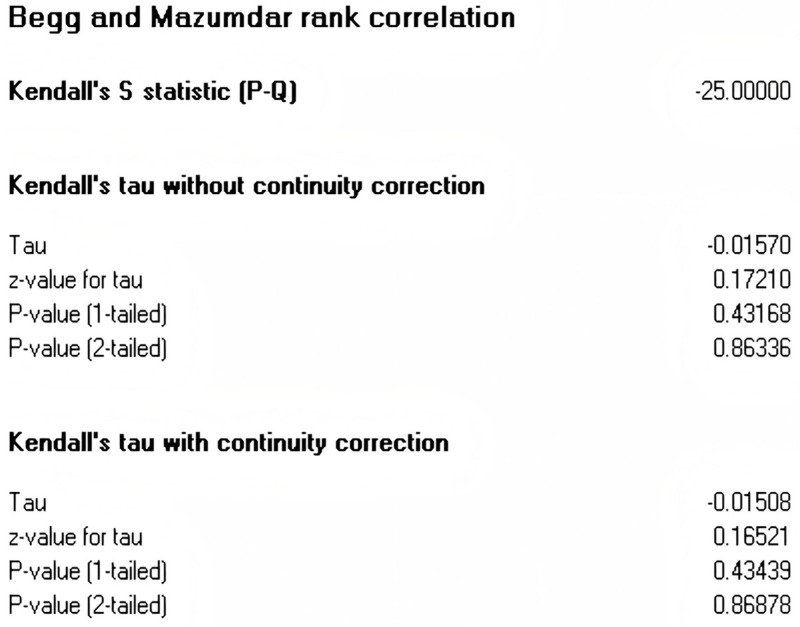
Kendall’s S statistic rank correlation.

**Table 1 behavsci-15-00342-t001:** Coding Results.

Studies Included		Journal Name	Sample Size	Participants	Emotion Regulation Strategies	Teachers’ Emotions	Correlation
							Coefficient
							Value
1	([43])	Educational Psychology	1281	Primary and secondary teachers in China	surface acting	emotional labor	0.51
					deep acting	emotional labor	0.56
					deep acting	job satisfaction	0.28
					deep acting	job satisfaction	0.17
					deep acting	job satisfaction	0.19
2	([4])	Stress and Health	2022	Middle and high school teachers in Croatian	deep acting	emotional and psychological traits	0.2
					deep acting	emotional and psychological traits	0.28
3	([44])	Teaching and Teacher Education	1115	Primary school teachers in China	surface acting	emotional labor	−0.64
					surface acting	teaching efficacy	−0.14
					surface acting	teaching efficacy	−0.16
					surface acting	teaching efficacy	−0.24
					deep acting	emotional labor	−0.01
					deep acting	teaching efficacy	0.19
					deep acting	teaching efficacy	0.11
					deep acting	teaching efficacy	0.12
4	([26])	Work (Reading, Mass.)	355	Preschool teachers in China	deep acting	emotional labor	0.38
					deep acting	emotional and psychological traits	0.14
					deep acting	emotional and psychological traits	0.13
					deep acting	emotional and psychological traits	0.17
					deep acting	emotional and psychological traits	0.13
					surface acting	emotional labor	−0.18
					surface acting	emotional and psychological traits	−0.24
					surface acting	emotional and psychological traits	−0.22
					surface acting	emotional and psychological traits	−0.24
					surface acting	emotional and psychological traits	−0.2
5	([27])	International Journal of Educational Management	322	Secondary teachers in Pakistan	surface acting	emotional and psychological traits	0.52
					surface acting	emotional labor	0.16
					surface acting	job satisfaction	0.29
					deep acting	emotional labor	−0.14
6	([1])	Stress and Health	659	Chinese teachers	deep acting	emotional labor	0.08
					deep acting	self-consciousness	0.2
					deep acting	self-consciousness	0.19
					deep acting	emotional labor	0.19
					surface acting	self-consciousness	0.14
					surface acting	self-consciousness	0.09
					surface acting	emotional labor	0.11
					surface acting	self-consciousness	0.08
7	([7])	System	594	Chinese teachers	surface acting	language and creativity	−0.83
					surface acting	emotional and psychological traits	−0.11
					surface acting	emotional and psychological traits	−0.21
					surface acting	emotional and psychological traits	−0.13
					surface acting	emotional labor	−0.18
					surface acting	emotional and psychological traits	−0.19
8	([21])	Teaching and Teacher Education	62	Chinese teachers	surface acting	emotional and psychological traits	−0.17
					surface acting	job satisfaction	−0.22
9	([42])	BMC Psychology	450	Chinese teachers	emotion regulation	emotional and psychological traits	0.57
10	([45])	Environ. Res. Public Health	1115	Chinese teachers	reappraisal	job satisfaction	0.16
					suppression	job satisfaction	−0.13
11	([46])	The Language Learning Journal	191	University teacher in Turkey	reappraisal	social belongingness	0.46
					reappraisal	social belongingness	0.44
					reappraisal	social belongingness	0.33
					reappraisal	social belongingness	0.36
					suppression	social belongingness	0.12
					suppression	social belongingness	0.16
					suppression	social belongingness	0.22
					suppression	social belongingness	0.13
12	([22])	Porta Lins.	329	All levels teacher in China	emotion regulation	language and creativity	0.77

**Table 2 behavsci-15-00342-t002:** Heterogeneity test.

Number of Effect Size	Effect Size (Point Estimate)	95% CI Lower Limit	95% CI Upper Limit	Z-Value	*p*-Value	Q-Value	df (Q)	*p*-Value (Q)	I^2^	Tau^2^	Standard Error	Variance	Tau
57	0.086	0	0.17	1.95	0.051	3934.683	56	0.000	98.598	0.108	0.025	0.001	0.328

**Table 3 behavsci-15-00342-t003:** Results of effect size analysis.

Emotion Regulation Strategies	Teachers’ Emotion	k	r	*p*	95%CI (LL, UL)	Heterogeneity			T	T^2^
						Q	P_Q_	I^2^		
Surface acting	Emotional and Psychological Traits	10	−0.131	0.099	(−0.250, 0.022)	178.859	0.000	94.968	0.215	0.046
	Emotional labor	6	−0.025	0.832	(−0.463, 0.382)	1085.184	0.000	99.539	0.563	0.317
	Job satisfaction	2	0.205	0.850	(−0.429, 0.506)	13.409	0.000	92.542	0.353	0.125
	Language and Creativity	1	−0.830	0.000	(−0.853, −0.803)	0.000	1.000	0.000	0.000	0.000
	Teaching Efficacy	3	0.103	0.000	(0.060, 0.147)	1.391	0.499	0.000	0.000	0.000
	Total	22	−0.076	0.280	(−0.281, 0.083)	2116.483	0.000	99.008	0.442	0.195
Deep acting	Emotional and Psychological Traits	6	0.215	0.000	(0.130, 0.242)	18.477	0.002	72.939	0.059	0.003
	Emotional labor	6	0.222	0.130	(−0.057, 0.417)	339.155	0.000	98.536	0.310	0.096
	Job satisfaction	3	0.213	0.000	(0.146, 0.280)	9.788	0.007	79.568	0.055	0.003
	Self-consciousness	2	0.195	0.000	(0.142, 0.246)	0.035	0.851	0.000	0.000	0.000
	Teaching Efficacy	3	0.140	0.000	(0.090, 0.189)	4.432	0.109	54.877	0.001	0.003
	Total	20	0.201	0.000	(0.119, 0.248)	394.051	0.000	95.178	0.148	0.022
Emotion regulation	Emotional and Psychological Traits	1	0.570	-	-	-	-	-	-	-
	Language and Creativity	1	0.765	-	-	-	-	-	-	-
	Total	2	0.668	0.000	(0.441, 0.827)	24.518	0.000	95.921	0.250	0.062
Reappraisal	Job satisfaction	1	0.160	-	-	-	-	-	-	-
	Social belongingness	4	0.397	0.000	(0.334, 0.458)	3.208	0.361	6.483	0.019	0.000
	Total	5	0.256	0.000	(0.207, 0.475)	33.556	0.000	88.080	0.162	0.026
Suppression	Job satisfaction	1	−0.130	-	-	-	-	-	-	-
	Social belongingness	4	0.156	0.000	(0.086, 0.225)	1.118	0.773	0.000	0.000	0.000
	Total	5	−0.014	0.262	(0.070, 0.252)	38.464	0.000	89.601	0.175	3.064

## Data Availability

The raw data supporting the conclusions of this article will be made available by the authors on request.

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
