# Peer review of "The Impact of Emotion Regulation Strategies on Teachers’ Well-Being and Positive Emotions: A Meta-Analysis"

_behavsci, 2025, doi:10.3390/bs15030342_

Round 1
Reviewer 1 Report
Comments and Suggestions for Authors
1. The study is well documented, organised and presented.
2. Comment the high heterogeneity between studies and how that helps or not generalization.
3. You could be more specific on the Boolean logic across multiple databases. Please indicate the exact terms used or procedures.
4. In the discussion section, it seems to me too general. It should be more specific and focus more on the actual results of the study.
5. You test publication bias and reserve to it a large section, but you hardly say anything in the discussion section and how that could influence the results.
6. Although you refer to deep acting and surface acting, how could other emotional regulation strategies help teachers better such as cognitive reappraisal or suppression?
Reviewer 2 Report
Comments and Suggestions for Authors
1.-Introduction
1.1.-I think it would be important to move the objective (lines 135 to 146) to the end of the introduction, after line 186.
Methodology
3.1.- I think that the flowchart should be revised, because in the section ‘items after eliminating duplicates’, it should not give 65, but many more. This can create confusion when reading it.
4. Results
4.1. Lines 447 and 448 state that the effectiveness of emotional regulation strategies in the teaching profession may be highly context-dependent. I think it would be appropriate to make explicit what is meant by context: type of school, school climate, classroom climate, level of teaching,...
Reviewer 3 Report
Comments and Suggestions for Authors
Thank you for the opportunity to review your paper.
This meta-analysis makes a valuable contribution to understanding emotional regulation strategies in teaching. The study's primary strength lies in quantifying the differential impacts of deep acting versus surface acting on teacher well being, providing empirical support for promoting deep acting strategies in professional development.
While the meta-analysis makes significant contributions to understanding emotional regulation in teaching there are some aspects of the paper that could be refined to strengthen the papers overall impact and utility.
Focus of the paper: The current paper opens with general statements about teaching and emotions but misses an opportunity to establish urgency and contemporary relevance. the introduction could be restructured to foreground the intensifying demands for emotional regulation in modern teaching environments. You could consider:
- establishing a clear problem statement early in the introduction (before line 21). You could include specific evidence of teacher burnout, rising attrition and intensifying emotional demands in contemporary education.
- strengthen the rationale for emotional regulation strategies. The current text in lines 28-31 mentions professional norms but doesn't adequately explain why understanding these strategies is crucial now.
- strengthen the research gap. While the text notes on lines 135-138 that 'significant gaps remain', it could explicitly state how understanding effective emotional regulation strategies could address current educational challenges. The gap could emphasise the practical implications: how better understanding of strategies could improve teacher retention as an example.
Theoretical Framework: The paper builds a solid theoretical foundation through careful examination of emotional labor theory and regulation concepts. To strengthen this aspect of the paper you may like to consider:
- inclusion of complementary current research for seminal works. The theoretical framing relies heavily on data sources, particularly evident in the citations on page 2 where Hargreaves (1998) and Gross (19980 form foundational arguments. The age of these work necessitates the inclusion of complementary current research or highlight a significant gap in current research.
- consider removal or realignment of concepts discussed within the literature review. The literature review in lines 89-104 introduces concepts about student learning processes that, while interesting, are not reflected in the study's findings. This section should ether be removed or better integrated with the study's actual outcomes. The theoretical discussion about language teachers specifically (lines 129-133) seems tangential to the broader focus of the meta-analysis and could be removed as it dilutes the papers focus on general teacher emotional regulation
- consider the inclusion of literature on the modern educational context. The theoretical foundation would benefit from incorporating contemporary challenges in education to address how contemporary educational challenges intensify teachers' emotional regulation demands. For example: teachers now navigate heightened emotional complexity, responding to increased student anxiety and mental health concerns. Additionally, diverse classroom environments demand more sophisticated emotional regulation as teachers work across various student needs. (Schonert-Reichl, K. A. (2017). Social and emotional learning and teachers. The future of children, 137-155.)
Research Design: The systematic met-analytic methodology demonstrates clear rigor in study selection and analysis. The inclusion and exclusion criteria are well developed in Section 3.2, where specific parameters for study selection are thoroughly detailed. To improve this section you may like to consider:
- restructuring the beginning of your methodology section, beginning with Section 3 (line 189). The PRISMA protocol should be foregrounded rathan than starting directly with search methods.
- consider explicitly delineating between each research phase. For example within Section 3.1 (lines 189-199) you may consider orientating the reacher to your research phases which would also be enhanced by numbering your phases: 1. systematic search and identification, 2 screening and eligibility assessment, 3. data extraction and quality assessment.
- consider strengthening the database selection rational in lines 190-191. You could expand this section beyond simply naming the databases. This section could justify why these specific databases were chosen by explaining their coverage strengths - for instance, WOS was selected for high impact peer-reviewed journals, EBSCO's PsycINFO and ERIC for educational psychology coverage, JSTOR for historical depth, and ProQuest for international research. This type of expansion would demonstrate a more rigorous and thoughtful approach to the systematic review process and help readers understand the comprehensiveness of the literature search.
- the coding procedures outlined in Section 3.4 (lines 271-276) could consider specific inter-coder reliability measures, coding scheme development and resolution procedures for coding disagreements.
- the assessment quality process is notably absent from the methodology. A new subsection should be added after Section 3.3 detailing how quality was evaluated using standardised assessment tools, particularly given this is a key component of PRISMA guidelines.
Discussion: The discussion would need to be reviewed to connect findings to contemporary challenges, if the positioning of the paper is refined to include modern educational demands.
Once again, thank you for the opportunity to review the paper. The suggested refinements are intended to strengthen what is already a valuable contribution to understanding emotional regulation in education.
Round 2
Reviewer 3 Report
Comments and Suggestions for Authors
Dear Authors,
Thank you for your considered responses to the initial review comments. The revised manuscript demonstrates some improvements in addressing the feedback. You strengthened the introduction with contemporary contextual data, enhanced the the theoretical framework with current research, articulated the research gap more explicitly, restructured the methodology to align more closely with PRISMA guidelines and connected findings to modern educational challenges. I note that information describing the PRISMA process for assessing the quality of each paper was missing which would strengthen the methodological rigour, however this single omission does not detract from the overall quality of the paper.
While minor improvements could still be made to the quality assessment process and coding procedures, you have successfully addressed the major concerns raised in the review, resulting in a more robust and relevant contribution to understanding emotional regulation strategies in teaching.
I look forward to reading the published version.
Comments on the Quality of English Language
The English language quality throughout the manuscript is generally good, with clear expression and appropriate academic tone, though a few minor grammatical and typographical issues could be addressed future copyediting.
For example: Line 506: "in certain contexts, surface acing". Should be "surface acting"